# Influence of Sample Matrix on Determination of Histamine in Fish by Surface Enhanced Raman Spectroscopy Coupled with Chemometric Modelling

**DOI:** 10.3390/foods10081767

**Published:** 2021-07-30

**Authors:** Sanja Vidaček Filipec, Davor Valinger, Lara Mikac, Mile Ivanda, Jasenka Gajdoš Kljusurić, Tibor Janči

**Affiliations:** 1Faculty of Food Technology and Biotechnology, University of Zagreb, Pierottijeva 6, 10000 Zagreb, Croatia; svidacek@pbf.hr (S.V.F.); jgajdos@pbf.hr (J.G.K.); tjanci@pbf.hr (T.J.); 2Center of Excellence for Advanced Materials and Sensing Devices, Ruđer Bošković Institute, Bijenička c. 54, 10000 Zagreb, Croatia; lmikac@irb.hr (L.M.); ivanda@irb.hr (M.I.)

**Keywords:** histamine, fish, matrix effect, SERS, silver colloid, PLS regression, principal component analysis, artificial neural network, rapid methods

## Abstract

Histamine fish poisoning is a foodborne illness caused by the consumption of fish products with high histamine content. Although intoxication mechanisms and control strategies are well known, it remains by far the most common cause of seafood-related health problems. Since conventional methods for histamine testing are difficult to implement in high-throughput quality control laboratories, simple and rapid methods for histamine testing are needed to ensure the safety of seafood products in global trade. In this work, the previously developed SERS method for the determination of histamine was tested to determine the influence of matrix effect on the performance of the method and to investigate the ability of different chemometric tools to overcome matrix effect issues. Experiments were performed on bluefin tuna (*Thunnus thynnus*) and bonito (*Sarda sarda*) samples exposed to varying levels of microbial activity. Spectral analysis confirmed the significant effect of sample matrix, related to different fish species, as well as the extent of microbial activity on the predictive ability of PLSR models with R^2^ of best model ranging from 0.722–0.945. Models obtained by ANN processing of factors derived by PCA from the raw spectra of the samples showed excellent prediction of histamine, regardless of fish species and extent of microbial activity (R^2^ of validation > 0.99).

## 1. Introduction

Histamine fish poisoning is an allergy-like form of food poisoning caused by consumption of different foods containing high concentration of histamine. In fish products, histamine is formed due to bacterial decarboxylation of free histidine caused by poor handling, i.e., time—temperature abuse of fish [1]. Majority of histamine fish poisoning incidents are associated with consumption of fish species containing high amount of free histidine in their tissues, such as tuna, mackerel, mahi-mahi, sardines and anchovies, although species with lower amount of free histidine, e.g., salmon and swordfish, are often reported as causative agents in histamine fish poisoning incidents [2,3]. Regulatory limits for histamine content of fishery products are established in many countries and range from 50 mg/kg of fish in the USA [4] to 100 mg/kg of fresh fish or 200 mg/kg of enzyme matured products in EU [5].

Traditional methods for histamine analysis and testing include high-performance liquid chromatography (HPLC), gas chromatography (GC), thin layer chromatography (TLC), enzymatic-linked immunosorbent assay (ELISA) and fluorometric methods [2,6] and are often time consuming, require expensive equipment and skilled laboratory personnel. Although intoxication mechanisms, control strategies and hazardous products are well known and documented, histamine is still the far most common cause of health problems associated with seafood [7,8,9] which leads to a conclusion that simple and rapid methods for histamine testing are required, in order to ensure safety of fish products in global trade. This is, in addition, confirmed by large number of recent scientific papers dealing with improvement of traditional histamine testing methods [10,11,12,13,14,15] or development of novel approaches such as different sensors [16,17,18,19,20] and application of different spectroscopic methods [21,22,23,24,25,26,27].

Among various spectroscopic techniques examined for rapid food analysis, Surface Enhanced Raman Spectroscopy (SERS) has gained significant interest due to high sensitivity, short analysis time and suitability for on-field analysis [28]. SERS combines Raman spectroscopy, which obtains molecular “fingerprint” of sample through measurement of inelastic light scattering, with nanotechnology. Enhancement of Raman signals is accomplished through electromagnetic and chemical mechanisms, when molecules of analyte are deposited onto metallic substrate with nanoscale surface roughness (SERS substrate) [29]. Electromagnetic enhancement originates through strong electromagnetic field generated by localized surface plasmons excited on metallic nanostructure by incident light beam while chemical enhancement arises from increased polarizability of molecule adsorbed onto SERS substrate [30,31]. In certain cases, combination of these enhancement mechanisms generates Raman enhancement factor up to order of 10^14^, enabling detection of single molecule of analyte [32,33,34]. However, application of SERS, for analysis complex matrices such as food, still faces challenges in terms of enhancing sensitivity and selectivity, reduction of matrix interferences, non-destructive sampling techniques and in-situ application. In addition, interpretation of spectral readings obtained by SERS often isn’t straightforward and requires use of different chemometric modelling tools to produce satisfactory results.

In recent years, several studies dealing with SERS detection and quantification of histamine in fish products have been published [21,22,26,27,35]. Majority of these studies were conducted on single fish species or on fish samples spiked with different concentration of histamine. Although this approach is valid for method development, it does not give much information on influence of sample matrix on method performance and overall applicability of such method for real-life applications. Sample matrix can differ significantly among different fish species. In addition, it may vary depending on extent of microbial activity within the sample since microbial activity isn’t restricted solely to histamine production but includes degradation of majority of components present in fresh fish and formation of wide array of chemical components. Since our previous study showed significant influence of sample matrix on performance of developed SERS method for determination of histamine in fish [22,36], this work was focused on testing the method on different fish species and on fish samples containing naturally formed histamine. In addition, different chemometric tools including Partial Least Squares Regression (PLSR) and Artificial Neural Networks (ANN) coupled to Principal Component Analysis (PCA) were employed to obtain prediction models (quantitative and qualitative) and test method performance with different fish samples.

## 2. Materials and Methods

### 2.1. Chemicals and Reagents

Silver nitrate (AgNO_3_, 99.9%), trisodium citrate (Na_3_C_6_H_5_O_7_×2H_2_O, 99.0%), sodium hydroxide (NaOH, 98.0%) and 1-butanol (C₄H₉OH, 99.5%) were purchased from Kemika d.d. (Zagreb, Croatia). Perchloric acid (HClO_4_, 65.0%) and sodium chloride (NaCl, 99.0%) were purchased from Carlo Erba Reagents (Milano, Italy). Sodium borohydride (NaBH_4_, 99.0%) was purchased form Acros Organics (Morris Plains, NJ, USA). Histamine dihydrochloride (C_5_H_9_N_3_ × 2HCl, 99.5%) was purchased from Applichem GmbH (Damstadt, Germany). All chemicals were of analytical grade and were used without further purification. High-purity water with a resistivity of 18 MΩcm was used in all experiments.

### 2.2. Fish Samples

Fresh samples of bluefin tuna (*Thunnus thynnus*) and bonito (*Sarda sarda*) were collected from the local market and transported to the laboratory. Skin, bones and viscera were removed and muscle tissue was minced using a stick blender. The histamine content of the fresh samples was analyzed by the reference HPLC method and the analysis confirmed that the fresh samples did not contain histamine. Minced samples of each species were divided into two groups. The first group (“calibration samples”) was spiked with histamine dihydrochloride to obtain final concentrations of 0, 25, 50, 75, 100, 125, 150, 175, 200, 250 and 300 mg/kg fish. The second group (“real samples”) was stored at room temperature to allow natural formation of histamine. Two samples were taken every 6 h and frozen until further analysis by HPLC and SERS method.

### 2.3. Determination of Histamine by Reference HPLC Method

Samples were analyzed according to the method of Malle et al. [37] officially recommended by the EU authorities. Briefly, 500 µL of internal standard (imidazole, 5 mg/mL) was added to 5.0 g of sample and homogenized in 50 mL of 0.4 M HClO_4_. The homogenate was centrifuged (3500× *g*/3 min) and 100 µL of the supernatant was mixed with 200 µL of saturated Na_2_CO_3_ solution and 500 µL of dansyl chloride solution (5 mg/mL in acetone). The prepared mixture was kept in a water bath at 60 °C for 60 min. Then, 100 µL of L-proline (100 mg/mL) was added and the mixture was left in the dark for 30 min. 500 µL of toluene was added and mixed with a vortex. After separation, 200 µL of the upper layer was transferred to a vial, evaporated in a stream of nitrogen until dry and dissolved in 200 µL of acetonitrile. The prepared samples were analyzed on an Agilent 1200 series liquid chromatograph using a LiChrospher C-18 column, 5 µm, 250 × 4.6 mm (Merck, Darmstadt, Germany) at 25 °C. Solvent A (water: acetonitrile, 40:60) and solvent B (acetonitrile) were used for gradient elution at a flow rate of 1 mL/min. Histamine was detected with DAD detector at a wavelength of 254 nm.

### 2.4. Preparation of Citrate Reduced Silver Nanoparticles (AGC)

The AGC colloid solution was prepared using the chemical reduction method according to the slightly modified method of Lee and Meisel [38]. Briefly, 90 mg of AgNO_3_ in 500 mL of miliQ H_2_O was poured into a four-necked glass flask and heated to boiling with constant stirring using a glass stirrer in an oil bath (120 °C) under reflux and nitrogen bubbling. 50 mL of 1% trisodium citrate was added rapidly to the boiling solution and stirred vigorously throughout. The solution was refluxed at 120 °C for 90 min and then removed from the oil bath and stirred until it cooled to room temperature.

### 2.5. SERS Measurements

Samples were prepared according to our previously developed method [22], which included homogenizing 5.0 g of minced fish muscle in 50 mL of 0.4 mol/L perchloric acid using the Ultra Turrax T-18 (IKA-Labortechnik, Staufen, Germany) laboratory homogenizer for 2 min at maximum speed and filtering through Whatman No. 4 filter paper. Then, 2 mL of the filtered extract was transferred to a glass test tube with a screw cap and 0.4 mL NaOH (5 mol/L), 1.4 g NaCl, and 2 mL of 1-butanol were added. The mixture was shaken on a laboratory shaker for 10 min at 70 rpm and allowed to stand for a few minutes to allow the layers to separate. 100 µL of the top butanol layer was transferred to a 1.5 mL capped micro centrifuge tube and evaporated under a stream of nitrogen at 60 °C until completely dry. 80 µL of AGC colloid, 10 µL of water and 10 µL of aggregating agent (0.23 mol/L NaBH_4_) were added to the dried sample, vortexed for 30 s, placed in a glass capillary (25 × 2 mm) and positioned in the macro chamber of the Raman instrument.

### 2.6. Instrumentation

Raman spectra were recorded using the Jobin Yvon T64000 triple Raman spectrometer (Horiba, Oberursel, Germany) in subtractive mode equipped with an argon ion laser with an excitation wavelength of 514.5 nm. The laser beam was focused with a 100 mm lens in 90° geometry and the laser power on the sample was approximately 60 mW. Raman scattering experiments were performed at room temperature. Spectra were recorded with 30 s accumulation and three replicates unless otherwise stated. The spectrometer was calibrated using the Raman spectrum of the transversal optical mode at 520.7 cm^−1^ of an undoped silicon wafer with (111) surface orientation.

### 2.7. Spectral Analysis and Chemometric Modelling

#### 2.7.1. Partial Least Squares Regression

In developing the models, specific regions of the SERS spectra were selected, all based on previous literature data examining the vibrational spectra of histamine, i.e., the intensity of the specific vibrational band of the histamine molecule [39] and four different wavelength combinations were selected. Thus, model M1 was developed in the wavelength range: 1140–1618 cm^−1^; for model M2, two ranges were chosen: 1240–1350 cm^−1^ & 1500–1600 cm^−1^; for model M3, six significant wavelengths were selected: 1264, 1268, 1304, 1320, 1437 & 1570 cm^−1^; and for the last model, M4, three following wavelengths were selected: 1268, 1320 & 1570 cm^−1^.

PLSR models [24] were developed on set of calibration samples spiked with histamine concentrations ranging from 0–300 mg/kg and containing 2048 wavelengths (1068.7–1618.1 cm^−1^) of each of the 479 SERS spectra. Prior to modeling, various preprocessing methods were applied to the SERS spectra including (i) Normalize (N); (ii) Baseline (B); (iii) Standard Normal Variate (SNV); smoothing methods such as (iv) Moving Average (MA); (v) Gaussian filter (GF); (vi) Median filter (MF); (vii) Savitzky-Golay Smoothing (SG_Sm) as well as combinations of two preprocessing methods such as (viii) Baseline coupled with the Savitzky-Golay derivation (S-G, B+dSG) & (ix) SNV combined with the Savitzky-Golay derivation (SNV+dSG). The efficiency of the model was evaluated using the root mean square error of cross validation (RMSEC) and the coefficients of determination (R^2^).

The obtained PLSR calibration models were applied to SERS spectra of real samples for all fish samples together (*n* = 196) and separately for bluefin tuna (*Thunnus thynnus*) (*n*_T_ = 160) and bonito (*Sarda sarda*) (*n*_B_ = 36). Data for all PLSR models were randomly divided into 60:40 ratio for calibration and validation. The efficiency of histamine concentration prediction was evaluated using the error of estimation (RMSEV) and R_v_^2^ (representing predictive ability). The Unscrambler X software (CAMO Software, Oslo, Norway) was used for preprocessing and modeling data in developing PLSRs.

#### 2.7.2. Principal Component Analysis

Although often used to detect similarities/differences between samples and to detect adulteration of samples [40], in our case, PCA was used to shorten the data matrix because it has the ability to extract important information from the data matrix and express it as factors (principal components). From the eigenvalues, the signifficant factors were selected and later used as input for the ANN modeling. Raw spectra without any preprocessing method were used for PCA analysis using Statistica v.13.0 software (StatSoft, Tulsa, OK, USA).

#### 2.7.3. Artificial Neural Networks

ANNs in combination with PCA were used to predict histamine concentration in different fish samples. The first 10 factors obtained by PCA were used as input for ANNs with hidden layer consisting of 3–14 neurons and histamine concentration as output. In addition, random separation of data into different ratios for training, testing, and validation was tested so as not to overfit or underfit ANNs. In all cases, random separation of data into 60:20:20 ratios for training, testing, and validation proved to be optimal. From the multiple layer perceptron networks that were developed in Statistica v.13.0 software (StatSoft, Tulsa, OK, USA), the ANNs presented in the results were selected based on the highest R^2^ values for training, testing, and validation along with the lowest root mean square errors.

## 3. Results and Discussion

### 3.1. Histamine Content and Spectral Features of Fish Samples

Reference HPLC analyses of sample sets containing naturally formed histamine showed significant formation of histamine in the samples during storage at room temperature. Since our SERS method has been shown to be suitable for the determination of histamine in the concentration range 0–300 mg/kg [22], samples with histamine contents above 300 mg/kg were excluded from further analysis. The histamine content of the real samples used for method testing is shown in Table 1.

From the SERS spectra of the histamine spiked calibration samples shown in Figure 1, it can be concluded that the characteristic SERS bands of histamine at 1264, 1320 and 1570 cm^−1^ and the citrate band at 1437 cm^−1^ [22,36] are clearly visible in the spectra of both fish species. Histamine SERS bands are of higher relative intensity and regular shape in the spectra of tuna sample, which can be explained by the intrinsic characteristics of the different fish species. It is possible that the matrix of the bonito samples contains a higher amount of interfering substances, resulting in slight spectral shifts and partial masking of the visible histamine bands.

Comparison of SERS spectra of calibration and real samples reveals significant changes in their spectral characteristics. While the baseline of the spectra of the calibration samples is flat, the baseline of the real samples shows higher intensity, which is especially observed in the lower wavenumber region. Moreover, significant shifts and distortions of the histamine SERS bands can be observed in the spectra of both fish species, although this effect is more pronounced in the spectra of the bonito samples, which contain naturally formed histamine. The described effect is a consequence of decomposition of the sample matrix due to various factors which include microbial activity, activity of endogenous enzymes and chemical reactions resulting in the formation of various degradation products, which may exhibit higher fluorescence when present in the extract. In addition, the SERS bands of the degradation products contribute to the SERS spectra of the sample and may cause spectral shifts or mask the characteristic SERS bands of histamine.

### 3.2. PLSR Models

The primary objective of applying PLSR models was to quantitatively predict histamine concentration in real fish samples with naturally formed histamine based on the collected SERS spectra. From the results presented in Figure 2, it is evident that the quality evaluation parameters of quantitative prediction are best for the model where the Baseline method was used to preprocess the spectral data (Table 2). Therefore, the baseline preprocessing method was subsequently applied to SERS spectra used for other models (Table 3).

Despite good qualitative indicators, prediction is not equally successful for both fish species (Table 2) and in different ranges of spectral wavelengths (Table 3). Equal prediction efficiency for bluefin tuna samples was obtained with model M1 (1140–1618 cm^−1^) and model M2 (1240–1350 cm^−1^ & 1500–1600 cm^−1^), which achieved R_v_^2^ of 0.913 and 0.912, respectively. The best results for the bonito samples and the combined bonito and tuna samples were obtained with model M2, achieving R_v_^2^ of 0.722 and 0.786, respectively.

As expected, the best predictive capabilities were obtained for the M2 model, since it is based on spectral regions containing vibrational modes of medium and strong intensity of the histamine molecule, i.e., the 1255–1324 cm^−1^ and 1480–1570 cm^−1^ regions, as previously reported [22,36,39,41]. Less acceptable predictive capabilities are determined for the M3 and M4 models based on a fewer individual strong vibrational bands of the histamine molecule, indicating that weaker vibrational bands present in a broader spectral range contain information that contributes significantly to the overall quality of the PLSR model.

The comparison of PLSR model parameters presented in Table 3 shows that the effect of sample matrix has a significant impact on the predictive capabilities of the PLSR models, as the best overall results were obtained for histamine spiked calibration samples (R_c_^2^ = 0.945). The lower performance of the models applied to SERS spectra of real fish samples with naturally formed histamine can be explained by the microbial activity that leads to the decomposition of the fish sample and the formation of various degradation products in addition to histamine formation. Since the sample preparation procedure for SERS measurement involves partial removal of matrix components from the extract, the vibrational bands of the compounds present in the extract may partially mask the SERS signals of histamine and affect PLSR model performance. The matrix effect can also be observed when comparing the results obtained for different fish species. The best results were obtained for bluefin tuna samples (R_v_^2^ = 0.912), while bonito samples may contain higher amounts of interfering compounds, resulting in lower model performance (R_v_^2^ = 0.722). In the case of applying the PLSR model to combined tuna and bonito samples, the result obtained (R_v_^2^ = 0.786) is intermediate between the results obtained for the individual fish species due to lower model performance for predicting histamine concentration in bonito samples.

### 3.3. Artificial Neural Network Models

To predict histamine concentration in fish samples, ANN were used in combination with PCA. Our experience in modelling with ANN in combination with PCA [40,42,43] has shown that preprocessing of spectral data before PCA analysis can sometimes improve the final result in terms of model performance [43]. Since preprocessing large amounts of data is time-consuming, laborious and requires modeling experience, in this work PCA was performed using the raw SERS spectra of the fish samples in order to minimize the required time for overall data analysis and thus provide faster method.

Furthermore, PCA is able to cope with small baseline variations from spectrum to spectrum [44] and it can perform very well with variety of spectroscopic data that are related [45]. In this case, PCA was used primarily to compress the data matrix used as loading for ANN. For example, the data matrix for combined fish samples in model M1 (wavelength range of 1140–1618 cm^−1^) is 196 × 1792, where 196 is the number of fish sample spectra and 1792 is the number of wavelengths in the range of 1140–1618 cm^−1^. From the derived eigenvalues obtained by PCA, it is possible to determine the number of factors necessary for further analysis. In our case, the first 10 factors were responsible for 99.99% of all variations in the samples, which means that none of the important information was lost, compressing our data matrix from 196 × 1792 to 196 × 10. Therefore, the first ten factors generated by PCA were later used as inputs for ANN, and single output for ANN was histamine concentration.

The first ANN models were developed for calibration sample sets, for each fish species individually and for combined fish samples (bonito and tuna). Calibration of the combined fish samples was performed using data from both individual fish species in the same matrix. Different ANN models were tested to determine which data separation could be used for all further analyses to avoid overfitting or underfitting of ANNs. Among the different data separations tested (70:15:15; 50:30:20; 60:20:20; 70:20:10), the best results were obtained by a random data separation of 60:20:20 for training, testing and validation, respectively. The results of ANN model predictions of histamine concentration for common fish samples and for each species separately are shown in Table 4.

As shown in Table 4, the M1 model (1140–1618 cm^−1^) for calibration set of tuna samples showed the highest values in terms of R^2^ for validation (0.9945) with the lowest root mean square error of 9.3494, while the lowest R^2^ value for validation was obtained for the combined calibration sets of both species with a value of 0.9754 and an root mean square error of 20.4546. From the network architecture, it can be seen that the M1 model for tuna had 10 inputs which were 10 first factors derived from PCA, 13 neurons in the hidden layer and one output which was histamine concentration, while the M1 model for bonito and combined samples had 12 neurons in the hidden layer. For the calibration of the combined fish samples, the highest values of R^2^ for training, test and validation (0.9960, 0.9911 and 0.9906) were obtained with the M2 model (1240–1350 cm^−1^ & 1500–1600 cm^−1^) with the lowest errors of 7.8974, 11.2420 and 13.2226, respectively. Slightly lower values were observed for model M3 (1264, 1268, 1304, 1320 cm^−1^) and the lowest prediction of histamine was obtained for model M4 (1268, 1320, 1570 cm^−1^).

As for the results obtained for bonito samples, the highest R^2^ values were also obtained for model M2, followed by model M1, M3 and the lowest values for model M4. Models M1, M3 and M2 obtained for tuna samples all had R^2^ values for validation above 0.99, with M1 having the highest value of 0.9945. As in the previous cases, model M4 had the lowest R^2^ value. To further test whether combining calibration samples with the addition of real fish samples would yield similar results, ANN models were developed with the same separation of data (60:20:20).

The results presented in Table 5 show that the models for combined calibration and real tuna samples had higher R^2^ values for all tested models. In addition, model M2 showed the highest predictive ability of histamine concentration, followed by models M3 and M1. As for the models constructed for separate calibration samples, model M4 showed the lowest ability to predict histamine concentration. For bonito samples, the highest values were also obtained with model M2, followed by models M1 and M3, while model M4 showed the lowest quality.

To test the ability of the ANN models to predict histamine concentration in real fish samples, the data for the calibration sample sets were excluded and ANN models were constructed using SERS spectra of real fish samples for individual fish species as well as for combined samples (Table 6).

It is interesting to see from the results in Table 6 that the R^2^ values of all ANN models were higher than 0.9602. The M1, M2 and M3 models showed excellent prediction of histamine concentration in the samples in all cases, while the M4 model, which previously had R^2^ values below 0.8630 (Table 4 and Table 5), showed the lowest value of 0.9602 for combined bonito and tuna samples in this case. As was the case with the ANN models for the calibration sets and the combined calibration and real sample sets, the models showed the highest possibility of histamine prediction for tuna samples. For bonito samples, it can be argued that the sample size was too small for the application of ANN as only 36 samples were tested. Although ANN models are more accurate for large datasets [46], this test was conducted to test whether it would be possible to obtain good predictions on such a small dataset. It can be clearly seen that better results were obtained for the tuna dataset, which had 160 samples.

Comparing the results of ANN with PLSR models, it can be concluded that ANN in combination with PCA is the more suitable method for the determination of histamine in fish using SERS. This could be explained by the fact that the intensity changes of histamine SERS bands do not show a linear trend with histamine concentration, but are better described by the Langmuir adsorption isotherm, which describes the adsorption of histamine molecules on SERS substrate [22,36]. Since PLSR models are better suited for linear models, one might expect ANN models to be better suited for this type of predictions.

## 4. Conclusions

The results have shown that there is significant influence in certain wavelength ranges of Raman spectra that were studied for both tuna and bonito samples due to the differences between two species and the extent of microbial activity leading to the formation of histamine and decomposition of the samples.

The observed changes in the spectra of the samples negatively affected the predictive ability of the PLSR models, regardless of the different pre-processing methods applied to the spectra. The best results were obtained by baseline correction of the raw spectra for PLSR model built on combined spectral range (1240–1350 cm^−1^ & 1500–1600 cm^−1^), resulting in R_v_^2^ of 0.912, 0.722 and 0.786 for the prediction of histamine concentration in tuna, bonito and combined samples, respectively.

Processing of the raw spectra by PCA allowed the extraction of factors accounting for 99.99% of all variations in the samples and significant compression of the data matrix for ANN modelling. The obtained ANN models showed an excellent prediction of histamine, regardless of the fish species and the extent of microbial activity, and the best results were also obtained for the model built on combined spectral range (1240–1350 cm^−1^ & 1500–1600 cm^−1^) with an R_v_^2^ greater than 0.99 in all cases studied.

## Figures and Tables

**Figure 1 foods-10-01767-f001:**
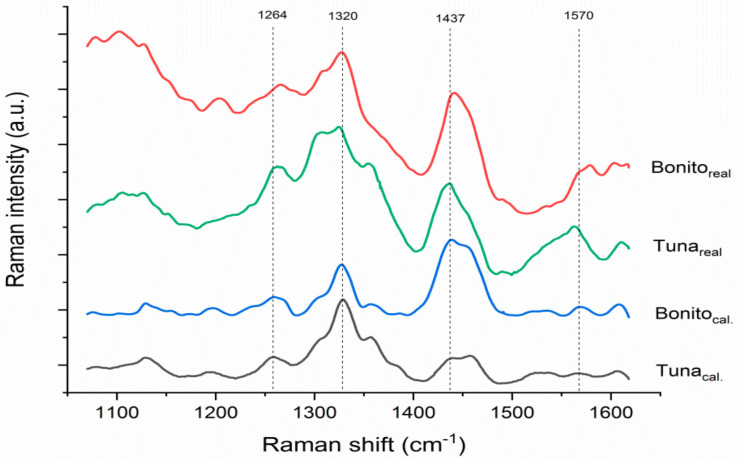
SERS spectra of calibration samples spiked with 75 mg/kg histamine and real samples T5 and B3 with similar histamine content.

**Figure 2 foods-10-01767-f002:**
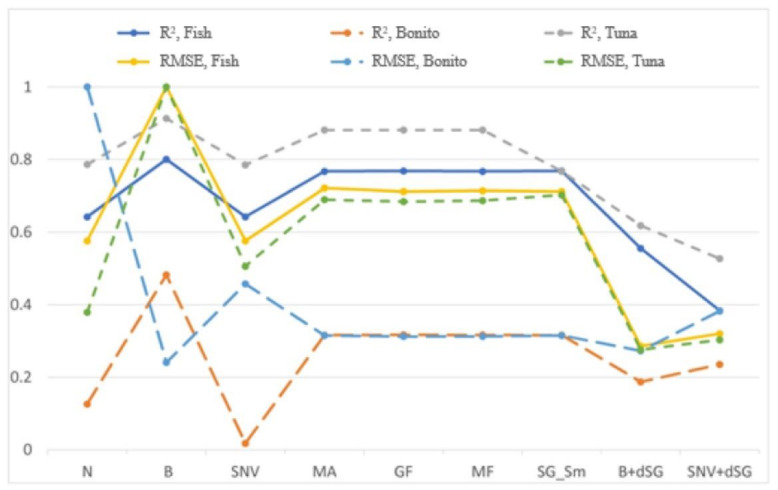
Validation results of PLSR equations for different input data, obtained by different preprocessing of SERS spectra of combined fish samples (Fish) and individual fish species (tuna, bonito) with corresponding root mean square errors (RMSE).

**Table 1 foods-10-01767-t001:** Histamine content of samples with naturally formed histamine determined by the reference HPLC method expressed as mean (*n* = 3) ± SD (tuna samples: T1–T7, bonito samples: B1–B5).

Sample	Histamine Content (mg/kg)	Sample	Histamine Content (mg/kg)
T1	8.6 ± 3.3	T7	248.1 ± 11.7
T2	9.6 ± 1.4	B1	33.6 ± 0.6
T3	23.0 ± 6.3	B2	41.1 ± 1.0
T4	33.3 ± 0.8	B3	76.1 ± 0.5
T5	81.8 ± 6.2	B4	136.2 ± 0.4
T6	173.7 ± 10.1	B5	184.9 ± 10.2

**Table 2 foods-10-01767-t002:** Results of model evaluation parameters (R^2^ and RMSE) for different preprocessing methods used for calibration data (model M1 in the range of 1140–1618 cm^−1^).

	Calibration	
Pre-Processing Method	R_c_^2^	RMSE_C_
Normalize, N	0.966	16.072
Baseline, B	0.978	13.169
Standard Normal Variate, SNV	0.984	11.127
Moving Average, MA	0.966	16.125
Gaussian filter, GF	0.971	14.969
Median filter, MF	0.967	15.904
Savitzky-Golay Smoothing, SG_Sm	0.966	16.129
Baseline + der. S-G, B+dSG	0.959	17.805
SNV + der. S-G, SNV+dSG	0.972	14.747
Pre-processing method	0.966	16.072

**Table 3 foods-10-01767-t003:** Parameters of PLSR models based on different spectral regions (M2: 1240–1350 & 1500–1600 cm^−1^; M3: 1264, 1268, 1304, 1320, 1437 & 1570 cm^−1^; M4: 1268; 1320, 1570 cm^−1^).

Model	Calibration		Fish		Bonito		Tuna	
	R_c_^2^	RMSE_C_	R_v_^2^	RMSE_V_	R_v_^2^	RMSE_V_	R_v_^2^	RMSE_V_
M1	0.978	13.169	0.800	237.741	0.482	52.578	0.913	242.606
M2	0.945	20.701	0.786	140.529	0.722	43.262	0.912	131.231
M3	0.764	42.842	0.691	67.589	0.3775	56.698	0.879	68.901
M4	0.504	73.525	0.409	68.045	0.4022	50.376	0.422	71.555

**Table 4 foods-10-01767-t004:** Artificial neural network models for histamine prediction of calibration sample sets for combined fish samples (Fish) and individual fish species (Tuna, Bonito).

Fish	Model	Network Architecture	Training Perf.	Test Perf.	Validation Perf.	Training Error	Test Error	Validation Error	Hidden Activation	Output Activation
Fish	M1	10-12-1	0.9975	0.9828	0.9754	6.3774	15.3578	20.4546	Tanh	Exponential
	M2	10-6-1	0.9960	0.9911	0.9906	7.8974	11.2420	13.2226	Tanh	Identity
	M3	10-10-1	0.9897	0.9614	0.9585	12.6903	23.3557	28.1661	Exponential	Logistic
	M4	10-9-1	0.8045	0.7117	0.6644	52.6201	58.9764	71.8243	Logistic	Logistic
Bonito	M1	10-12-1	0.9964	0.9955	0.9910	7.6300	9.0898	13.0946	Exponential	Logistic
	M2	10-5-1	0.9981	0.9954	0.9916	5.6370	9.2371	10.1250	Tanh	Exponential
	M3	10-7-1	0.9964	0.9939	0.9870	7.8028	10.8345	12.6816	Logistic	Exponential
	M4	10-8-1	0.9047	0.8928	0.8507	39.4613	44.2791	49.3624	Tanh	Exponential
Tuna	M1	10-13-1	0.9981	0.9953	0.9945	5.3512	9.2580	9.3494	Tanh	Exponential
	M2	10-11-1	0.9979	0.9908	0.9902	6.0746	11.6041	13.0318	Tanh	Logistic
	M3	10-6-1	0.9959	0.9946	0.9919	7.8603	9.7427	11.2788	Logistic	Exponential
	M4	10-13-1	0.9295	0.8648	0.8269	31.8624	47.3664	49.4626	Logistic	Exponential

**Table 5 foods-10-01767-t005:** Artificial neural network models for histamine prediction for calibration sample sets combined with real sample sets of each fish species.

Fish	Model	Network Architecture	Training Perf.	Test Perf.	Validation Perf.	Training Error	Test Error	Validation Error	Hidden Activation	Output Activation
Bonito	M1	10-10-1	0.9965	0.9906	0.9874	7.5415	11.8646	12.7808	Logistic	Exponential
	M2	10-7-1	0.9964	0.9926	0.9883	7.6440	10.7651	13.4342	Tanh	Exponential
	M3	10-6-1	0.9931	0.9839	0.9797	10.6157	15.4946	16.6087	Exponential	Exponential
	M4	10-5-1	0.9350	0.8755	0.8564	32.2163	42.3773	49.1198	Logistic	Logistic
Tuna	M1	10-10-1	0.9977	0.9933	0.9920	6.3520	8.8020	11.6935	Logistic	Exponential
	M2	10-12-1	0.9981	0.9931	0.9930	5.8671	9.0178	10.6760	Tanh	Exponential
	M3	10-10-1	0.9980	0.9933	0.9925	6.0089	9.0942	11.3809	Tanh	Exponential
	M4	10-11-1	0.9399	0.8698	0.8630	32.2323	39.1596	48.9777	Exponential	Tanh

**Table 6 foods-10-01767-t006:** Artificial neural network models for histamine prediction in real fish samples for combined fish samples (Fish) and individual fish species (Tuna, Bonito).

Fish	Model	Network Architecture	Training Perf.	Test Perf.	Validation Perf.	Training Error	Test Error	Validation Error	Hidden Activation	Output Activation
Fish	M1	10-7-1	0.9988	0.9937	0.9936	3.9688	8.8349	9.7735	Tanh	Identity
	M2	10-8-1	0.9993	0.9973	0.9935	2.9749	5.8620	9.6090	Exponential	Tanh
	M3	10-10-1	0.9978	0.9939	0.9935	5.4714	9.0641	10.4907	Logistic	Tanh
	M4	10-5-1	0.9831	0.9653	0.9602	15.1824	21.7015	23.3168	Logistic	Exponential
Bonito	M1	10-10-1	0.9990	0.9976	0.9955	2.4070	5.3407	5.8885	Logistic	Exponential
	M2	10-10-1	0.9992	0.9972	0.9958	2.4093	4.8894	8.1239	Exponential	Logistic
	M3	10-4-1	0.9996	0.9989	0.9713	1.6074	5.1973	16.2568	Logistic	Identity
	M4	10-9-1	0.9982	0.9698	0.9603	3.8024	17.4051	21.3636	Tanh	Logistic
Tuna	M1	10-6-1	0.9995	0.9990	0.9988	2.8103	3.6519	4.3021	Exponential	Logistic
	M2	10-12-1	0.9998	0.9990	0.9989	1.6327	3.6420	4.6434	Tanh	Exponential
	M3	10-11-1	0.9999	0.9996	0.9987	1.2763	2.3012	4.5641	Tanh	Identity
	M4	10-4-1	0.9959	0.9924	0.9912	7.7512	10.0073	12.8127	Tanh	Tanh

## Data Availability

Not applicable.

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
