# Peer review of "Influence of Sample Matrix on Determination of Histamine in Fish by Surface Enhanced Raman Spectroscopy Coupled with Chemometric Modelling"

_foods, 2021, doi:10.3390/foods10081767_

Round 1

Reviewer 1 Report

The manuscript by Tibor Janci et al presents the influence of sample matrix on determination of histamine in fish by Surface Enhanced Raman spectroscopy coupled with 3 chemometric modelling. After a complete and precise introduction of the objectives and a fairly well described materials and methods part, the results and discussion part is difficult to read and therefore less clear, so it needs some clarification.

  1. As mentioned by the author in the abstract and introduction (line 12-13 and 31-32), “Histamine fish poisoning is an allergy-like form of food poisoning caused by consumption of different foods containing high concentration of histamine“.

In Part 3 Results and discussion the author states (Line 203-205)  “ Since our SERS method has been shown to be suitable for the determination of histamine in the concentration range 0 - 300 mg/kg [22], samples with histamine contents above 300 mg/kg were excluded from further analysis “.

How can we be sure that the histamine concentrations studied (0-300 mg/kg) correspond to the threshold at which intoxication exists since the author does not provide any information on this element.

In the following, § 3.1 the results seem to show, as assumed by the author, that it is the microbial activity that leads to the decomposition of the sample matrix and the formation of various degradation products and thus contributes to the degradation of the SERS spectra (line 226- 230).

Moreover the author says line 245-247 “ Equal prediction efficiency for bluefin tuna samples was obtained with model M1 (1140-1618 cm-1) and model M2 (1240 - 1350 cm-1 & 1500 - 1600 cm-1)“ but we do not have the results of the M1 model.

Error line 250 : replace M by M3

Finally, the same remarks can be made about § 3.3

If the results show indeed a significant influence of the matrix effect on the spectral characteristics of the studied samples, it remains difficult to form an objective opinion of the contribution of the obtained ANN models on the prediction of histamine

Reviewer 2 Report

This paper describes the influence of sample matrix on determination of histamine in fish by SERS coupled with chemometric. The article is quite complete, it is of interest to the scientific community, the methods and statistics used are appropriate and the results are conveniently described. The work is well discussed and is supported by the references provided by the authors. The authors have a great knowledge of the subject as it is observed in the bibliography, and this work deepens even more in this field.

I consider that the article is appropriate to be published in Foods journal once the authors have made minor modifications to it.

Title: Capitalize each word according the format of the journal.

Lines 26, 176: Put a separation after and before “>”, “=”. Unify and apply to the entire document.

Lines 90, 99, 110, 125, ……: Capitalize each word according the format of the journal. Unify and apply to the entire document.

Line 95: (Morris Plains, NJ, USA)

Line 115: “µL” instead of “µl”.

Line 121: (Merck, Darmstadt, Germany)

References: The name of the journals must appear abbreviated according to the format of the journal.

Round 2

Reviewer 1 Report

Dear author, the answers are appropriate to the questions asked.